# Incidence Rates and Risk of Hospital Registered Infections among Schizophrenia Patients before and after Onset of Illness: A Population-Based Nationwide Register Study

**DOI:** 10.3390/jcm7120485

**Published:** 2018-11-27

**Authors:** Monika Pankiewicz-Dulacz, Egon Stenager, Ming Chen, Elsebeth Stenager

**Affiliations:** 1Focused Research Unit of Psychiatry, Department of Psychiatry, Kresten Philipsens Vej 15, 6200 Aabenraa, Denmark; Elsebeth.Stenager@rsyd.dk; 2The Department of Regional Health Research, University of Southern Denmark, Winsløwparken 19.3, 5000 Odense, Denmark; egon.stenager3@rsyd.dk (E.S.); ming.chen@rsyd.dk (M.C.); 3Focused Research Group of Neurology, Department of Neurology, Hospital of Southern Jutland, Sønderborg, 6200 Aabenraa, Denmark; 4Department of Clinical Microbiology, Hospital of Southern Jutland, 6400 Soenderborg, Denmark

**Keywords:** epidemiology, schizophrenia, infections, susceptibility, population-based

## Abstract

Infections in schizophrenia patients are associated with an increased premature mortality. However, our knowledge about the burden of infections in schizophrenia is scarce. The aims of this study were to (1) determine the prevalence of clinically important hospital registered infections in the period of five years prior to and five years after the diagnosis, (2) estimate the risk of infections before and after the schizophrenia diagnosis and, (3) evaluate the impact of comorbidity on the risk of infections in schizophrenia. Using combined data from Danish national registers, we sampled a cohort of all persons born in Denmark in the period 1975–1990 and obtained health-related records from 1995–2013. Occurrence patterns and the risk of infections were measured as annual incidence rates and incidence rates ratios, estimated using Poisson models. Medical conditions from the Charlson Index were considered as a measure of comorbidity. The analyses showed that schizophrenia patients had a significantly elevated risk of almost all types of hospital registered infections during the period of the study when compared to the controls. Comorbidity increased rates of infections by 176%. The results suggest that the risk of infections is elevated in the schizophrenia population and physical illness is an important risk factor.

## 1. Introduction

The current evidence indicates that schizophrenia is associated with an increased risk of infections. Previously, several studies have documented an elevated frequency of various infections both before and after the schizophrenia diagnosis [1,2,3,4,5,6,7].

The high correlation between a broad range of infections in childhood and adolescence and a risk of developing schizophrenia has been found in a recent longitudinal population-based study of Nielsen et al. [2]. The authors proposed two hypotheses as explanations of this finding: first, that exposure to infections may increase the risk of schizophrenia due to inflammatory responses affecting the brain, and second, which claims that elevated susceptibility to infections in schizophrenia patients may lie in genetic and environmental factors rooted in families. Positive associations between parental infection and schizophrenia in offspring have been reported in a former large cohort study from Denmark. Results from that study suggest that schizophrenia may be associated with familial susceptibility to developing infections [8].

The consequences of infections in schizophrenia might be very serious. There is a large body of evidence that infections are associated with unfavorable clinical outcomes in this group of patients, including higher rates of premature death, an elevated risk of acute respiratory, and other organ failures [9,10,11]. Today, mortality due to infections in schizophrenia seems to be well acknowledged [12]. Still, evidence regarding the burden of infections in this group of patients is scarce. Only the prevalence of chronic infections like HIV, hepatitis B and C, and tuberculosis is well established today [13].

A very recent study based on a large Danish population demonstrated a greater risk of hospital contacts due to infections overall both in the pre- and post-diagnostic period of schizophrenia. However, the types of infections have not been investigated in detail [14].

Further, the potential influence of medical comorbidity on the risk of infections in patients with schizophrenia has rarely been considered in previous studies. Underlying physical illness, smoking, and alcohol abuse have been recognized as factors with a strong impact on the risk of infections. All of these factors are highly related to schizophrenia, and a majority of them occur before the onset of schizophrenia [15,16,17].

Another aspect that needs to be addressed is whether the risk of infections may vary across the schizophrenia illness course. Three clinical studies reveal that psychosis is associated with an increased risk of urinary tract infections [18,19,20]. A recent study of Carson et al. demonstrated that persons with non-affective psychosis were more than three times more likely to experience urinary tract infections during hospitalization when compared to non-psychotic depressive patients. In parallel, studies assessing relationships between antipsychotic medications and pneumonia found that treatment with both first-generation antipsychotic drugs and second-generation antipsychotic drugs increased the risk of pneumonia, especially at the beginning of the treatment [21,22].

Considering all these findings, we propose that persons at the time around and after the diagnosis of schizophrenia could be more sensitive to infections due to the start of antipsychotic treatment, psychotic symptoms, stress, or functional decline. The current knowledge about the occurrence patterns of infections among schizophrenia persons before, around, and after diagnosis remains lacking.

The understanding of these problems is crucial primarily from a clinical point of view as infections account for the leading causes of premature death among schizophrenia patients [12,23]. Detailed knowledge about the risk of infections during schizophrenia illness could enable us to identify the time-window at which schizophrenia patients are at the highest risk of infections and then recognize the time when the implementation of preventive interventions would be most beneficial. Furthermore, patients with schizophrenia are known to have a high occurrence of multimorbidity and substance abuse before the onset of the illness. Thus, there is a need for improved knowledge about the susceptibility to infectious diseases in schizophrenia patients where the impact of time and comorbidity is considered. Finally, there is a demand for new information that can provide a better understanding of this complex problem of susceptibility to infections in schizophrenia patients.

To overcome the limitation of the current evidence and the approach of the previously mentioned queries, we have conducted a large-scale population-based study with the objective to examine the occurrence patterns and risk for infectious diseases in the period prior to, around, and after the schizophrenia diagnosis. Specifically, we wanted to (1) determine the prevalence of clinically important infections in the five years before and five years after the diagnosis, (2) estimate the risk of infections before and after the schizophrenia diagnosis and, (3) evaluate the impact of comorbidity on the risk of infections in schizophrenia. The time frame of five years before and after the first medical record with schizophrenia was chosen to capture the most vulnerable periods surrounding the onset of the illness (two years before and three years after) [24,25].

## 2. Materials and Methods

### 2.1. Data Sources

In Denmark, each Danish resident is assigned a unique personal identification number (The Danish Personal Identification number: CPR number) which allows for record linkage across the entire registration system [26]. For the use of the study we obtained sociodemographic and health-related data from the following registers: The Danish Civil Registration System [27], The Danish National Patient Registry [28], The Danish Psychiatric Central Research Register (PCRR) [29], The Danish Register of Causes of Death (DRCD) [30].

### 2.2. Study Population

We sampled a birth cohort of all persons born in Denmark between 1 January 1975 and 31 December 1990 and living in Denmark between 1 January 1995 and 31 December 2013 from The Danish Civil Registration System. Through linkage to the Danish Psychiatric Research Register, we identified all individuals with schizophrenia. Information derived from the Danish Psychiatric Research Register was available from 1969. Cohort members were identified with schizophrenia if they had any inpatient or outpatient contacts with a diagnosis of schizophrenia (the eighth revision of The International Classification of Diseases and Related Health Problems—ICD-8 code: 295; the 10th revision of The International Classification of Diseases and Related Health Problems—ICD-10 code: F 20), irrespective of other additional or previous diagnoses [31]. The first date of the first registered psychiatric contact with the schizophrenia diagnosis was defined as an Index date. In Denmark until 1994, the eighth revision of The International Classification of Diseases and Related Health Problems (ICD-8) was applied. After that, the ICD-10 was implemented. Each schizophrenia patient was matched by age, sex, and year of the first psychiatric contact with schizophrenia to 10 control subjects, who had been randomly selected from the birth cohort. The Case-Control Matching Program in analytics software SAS was used to automate the matching of controls with cases. Control subjects were assigned the same index date as their corresponding case. Applying the matching method, we derived from the study cohort 7852 schizophrenia cases and 78,520 matched controls. Follow-up was defined as five years before and five years after the index date (date of inpatient or outpatient contact with the schizophrenia diagnosis). Register data on the study population was collected over a period of 18 years from 1 January 1995 to 31 December 2013. To identify physical illness, we searched all inpatient and outpatient medical records with infection and a chronic medical condition. The hospital registered infection was defined as an inpatient or outpatient medical record with a diagnosis of infection (ICD-10 codes J00-B00). All ICD-10 codes with modification code for “suspected” and “not found” were excluded. Infections were additionally grouped according to the type of infection following stratification in ICD-10. (See Appendix A for specification). We considered medical conditions from the Charlson Index as measures of comorbidity, and we defined them by applying ICD-10 codes [32]. We added substance abuse and obesity to the index as those diagnoses were very relevant for the outcome (See Appendix A for specification). Medical diagnoses were coded as binary variables (no comorbidity, comorbidity) and recorded before the start and during each year of follow-up. Persons were identified with comorbidity when they had any inpatient or outpatient medical record at the given year or in previous years of follow up with any diagnoses from the Charlson Index.

### 2.3. Statistical Analysis

The characteristics of the study population were found by comparing schizophrenia patients and the control group on background variables using two-way tables and the chi-square test. Infection patterns in the schizophrenia group and the control group were measured as the annual gender-specific incidence rates of major infection and estimated using a Poisson regression model using SAS GENMOD procedure in the SAS version 9.4 (SAS Institute, the maintenance level—TS1M3, Carry, NC, USA). Confidence intervals for the rates were also calculated. Incidence rates were additionally grouped according to comorbidity. In this part of the study, conditions of nested case-control studies were applied using a matched case-control population as previously described. Risks were calculated as incidence rate ratios (IRRs) for all types of infection and infections overall applying conditions of the cohort study and using the whole birth cohort in analysis with schizophrenia as exposure. To control for the influence of time on the risk of infections, we included calendar-years in our statistical models. Calendar years were categorized as 1-year periods and grouped according to the time before and the time after the onset of schizophrenia (11 categorical time variables). All risk factors inclusive of time variables were analyzed in the Poisson regression model. Inpatient and outpatient medical records with infection was our dependent variable. The results were returned as incidence rates ratio, *p* values, and 95% Cl. All individuals were considered at risk of infection until the occurrence of the following events: death, migration, and study end. Additionally, we censored persons from the study after the occurrence of hepatitis and HIV as those infections might confound our results due to their impact on the risk of other infections [33]. All *p* values less than or equal to 0.05 were considered statistically significant. The study was approved by the Danish Data Protection Agency with the number: 2008-58-0035.

## 3. Results

### 3.1. Descriptive Measures

The characteristics of patients with schizophrenia and controls are shown in Table 1.

This study included 893,647 people followed up to a total of 15,994,882.75 person-years from 1995–2013. We identified 7852 patients with schizophrenia during the study period of whom 2834 (32.6%) had an infection. Reflecting the established burden of comorbid medical conditions associated with schizophrenia, persons with a schizophrenia diagnosis had substantially higher rates of physical illnesses with 3345 (42%) persons with co-morbid conditions contra 162,816 (18.3%) from the general population. This substantial discrepancy in our study could be a result of the high prevalence of substance and alcohol abuse among schizophrenia patients, 20.54% versus 3.34% from the general population. However, increased co-occurrence regarded nearly all types of medical conditions. The most common medical conditions in both groups were chronic pulmonary diseases, followed by obesity, and inflammatory bowel syndrome. The frequency of cancer among persons with schizophrenia was slightly, but not significantly, lower than in the general population, which is in line with several previous studies [34].

### 3.2. Gender-Specific Annual Incidence Rates of Hospital Registered Infections among Schizophrenia and Control Subjects with and without Comorbidity

Annual gender specific incident rates of hospital registered infections with or without comorbidity for case subjects and controls are illustrated in Figure 1 and Figure 2. The influence of age and sex were controlled by the matching process of the study. The finding of this study documented that comorbidity is an important predictor for susceptibility of infections for both schizophrenia subjects and controls. Both females and males with medical conditions or substance abuse were almost 3 times more likely to suffer from infections (IRR = 2.76; CI = 2.69–1.84; *p* < 0.0001) when compared to the control subjects without medical conditions. Schizophrenia was related to a 63% increased risk of hospital registered infections for both sexes (IRR = 1.63; CI = 1.58–1.68; *p* < 0.0001). With regards to gender, we found that females had a greater manifestation of hospital registered infections (IRR = 1.60; CI = 1.56–1.64; *p* < 0.0001) when compared to males. This increase in the risk of infections, attributed to the female sex and measured in our study could be determined by gynecological, urological, and the other types of infections which were strongly associated with the female gender and which frequently occurred in the study population.

### 3.3. Time-Dependent Risk of Hospital Registered Infections Overall and Stratified by Type of Infection

The most frequent infections among schizophrenia patients were: respiratory infections (mean of respiratory infections in schizophrenia = 0.14, SD = 0.60 vs. mean of respiratory infections in cohort members without schizophrenia = 0.07, SD = 0.59), skin infections (mean = 0.30, SD = 1.67 vs. mean = 0.11, SD = 0.68), and gastrointestinal infections (mean = 0.12, SD = 0.44 vs. mean = 0.07, SD = 0.34). Time-dependent analysis showed an equally increased risk of any infection for schizophrenia patients during the study period (IRR = 1.63; CI = 1.58–1.68; *p* = 0.0001). Subgroup analysis by type of infection was performed and measured both as annual incidence rates and incidence rate ratios. All of the conducted subgroup analysis revealed no substantial difference in the association between the time-window of follow-up and type of infection. Therefore, our final models calculated incidence rate ratios during the entire follow up time of the study. Incidence rates ratios for different types of infection are presented in Table 2.

After adjusting for comorbidity, we found 4 infections which had the highest positive association to schizophrenia: skin infections (IRR = 1.88; CI = 1.81−1.96; *p* < 0.0001), urological infections (IRR = 1.80; CI < 1.65–1.95; *p* < 0.0001), genital infections (IRR = 1.91; CI = 1.73–2.11; *p* < 0.0001), and tuberculosis (IRR = 2.35; CI = 1.69–3.27; *p* < 0.0001). The infection which had the lowest positive association to schizophrenia was other types (IRR = 1.17; CI = 1.10–1.23; *p* < 0.0001), and gastrointestinal infections (IRR = 1.24; CI = 1.17–1.33; *p* < 0.0001). In both based and adjusted models, age was controlled by the matching process of the study.

## 4. Discussion

### 4.1. Principal Findings

This study provides three main findings. First, after adjusting for several comorbidities the data showed equally increased risk of hospital registered infections compared to the general population both, in the pre-diagnostic and post-diagnostic period. Secondly, our findings demonstrated that, elevated susceptibility to infections was not driven by any specific infection. Finally, confirming the hypothesis that the physical illness contributes substantially to the increased risk, we found that comorbidity increased susceptibility for infections with 176% for both the general population and schizophrenia patients.

### 4.2. Time and Risk of Infections

After accounting for potential impact of the comorbidity, we have failed to find significant differences in the magnitude of the association between the period of premorbid schizophrenia and the risk of infections, or in the period after the schizophrenia diagnosis and the risk of infections. This resemblance of associations found in our study is in accordance with the previous Danish population-based study [14]. Our understanding of why individuals with schizophrenia experience higher risk of infections-both in the time before and after the diagnosis is limited. However, there are several possible explanations.

A wide range of factors that might have contributed to a greater susceptibility to infections have been found among schizophrenia patients several years prior to the onset of the illness. These include depressive symptoms, poorer social functioning, cognitive and motoric impairment, substance abuse, and smoking cigarettes [35,36]. A recent study, based on medical records from the primary health care in Denmark found, that people with schizophrenia had an increased frequency of contacts due to mental problems up to six years before the diagnosis [36]. 30% of schizophrenia patients had at least one psychiatric contact six years prior to the diagnosis. One month before the schizophrenia diagnosis, three out of four patients had psychiatric contact. Andersson and colleagues documented in a recent study that depressive conditions are significantly associated with an increased risk of infections [37]. Functional decline might also be expected to be associated with an increased risk for infections. Additionally, the current literature shows that the premorbid phase of schizophrenia is associated with an increased prevalence of physical conditions [17]. Numerous somatic illnesses have been reported, where any of them could compress the immune system and thus lead to an increased vulnerability to infections. Notably, malnutrition and anemia which are strongly correlated with vulnerability to infections were frequently observed prior to the schizophrenia diagnosis. Thus, our inability to correct for all above mentioned conditions could at least partly explain our findings. The other possible explanation could be that pathways underpinning the association between schizophrenia and infections arise from risk factors common for both diseases, as suggested previously by Nielsen et al. [14]. Prior studies reported a link between schizophrenia and the immune system. Genetic studies have found an association between schizophrenia and the immune system, where clinical studies showed various immunological alterations in the blood of schizophrenia patients both before and after the diagnosis [38,39]. Alterations of the immune system among neonates who later in life develop non-affective psychosis have been reported in several, but not all, studies [40,41,42]. Furthermore, studies among first degree relatives have indicated a familial susceptibility to infections in schizophrenia patients [8]. Similarly, familial factors were found to contribute substantially to the association between the infection and schizophrenia in another population-based study [3]. Thus, several unmeasured factors such as life style, social adversity, stress, or familial influences that this study does not account for, could mediate this association. The influence of shared familial factors is, however, purely understood, and findings from the few recently published studies do not support the hypothesis [43,44].

Findings regarding types of infections prior to the schizophrenia diagnosis are generally a reproduction of the previous findings from the large cohort Danish study [2]. However, there are some differences. For instance, we found low positive association between sepsis after adjustment for comorbidities in a contrast to high positive association found in the previous study. This finding has an important clinical implication as sepsis continues to be a major cause of death. Findings from our study suggest that comorbidity may account for an important predictor for sepsis in schizophrenia patients. After adjusting for confounders four infectious diseases with the highest positive association to schizophrenia have been found: urological infections, genital infections, skin infections and tuberculosis. These association remained highly positive in both the pre-diagnostic and post-diagnostic period. Increased susceptibility due to those infections could be related to the life style of the person as well as immune disabilities in general. The association between tuberculosis and schizophrenia has been documented in many previous studies and today persons with schizophrenia are recognized as high-risk groups for the disease [13]. The cause for this correlation is largely unclear, however, the impact of physical illness, substance abuse, and social factors have been proposed as possible explanations. Currently knowledge about other types of clinically important infections after schizophrenia diagnosis remains limited. A large population study based on English records revealed increased risk of pneumococcal disease among schizophrenia patients [5]. Similarly, data from a Finnish study showed that schizophrenia was associated with increased odds for pneumonia [6]. A large-scale study based on Danish registers documented association between severe mental disorders and elevated risk of hospitalization due to pneumonia, angina, and urinary tract infections [7].

Collectively, by using data form a large nationally representative sample we assessed frequency and risk of hospital recorded infections in schizophrenia across the illness course. In addition, we were able to explore the role of physical illness in this association. Our model was, however, unable to explain the bidirectional association between schizophrenia and infections suggesting that the other potentially risk factors should be considered in future research. Consequently, the results did not allow us to clarify the causality of the association. Nevertheless, this study might have a high relevance for clinicians, as this provides a novel knowledge regarding burden of different types of infections in schizophrenia patients. Today, evidence indicate that utilization of medical care in patients with schizophrenia is reduced and no guidelines regarding preventive efforts for infections exist for this group of patients. This is mainly due to the fact, that individuals with schizophrenia are not considered to be at an increased risk of infections. At presence, only few studies attempted to investigate the risk of infection in schizophrenia and there is a need for more data to confirm this hypothesis. In the context of lifelong health, mortality studies continually report elevated rates of mortality due to infections in this group of patients. Therefore, preventive interventions that might help reduce the prevalence of infections could made a difference on the survival rates among schizophrenia patients. A more detailed understanding of the role of different factors on the susceptibility to infections may be crucial for both preventive efforts but also better understanding of the relationship between infections and schizophrenia. Thus, there is a need for further research in this field which will help to clarify the mechanisms mediating this association.

### 4.3. Strengths

Our data provide, for the first time, detailed information about the occurrence of the clinically important hospital recorded infections across the schizophrenia illness course. Additionally, we adjusted in our analyses for important confounders like comorbidities and substance abuse. Using this large-scale population-based approach, we minimized the selection biases, as all data were obtained from public hospital records. In context of representability, it is important to mention that Denmark provides free healthcare services to all citizens and all secondary healthcare contacts to hospital-based specialty clinics are recorded in registers. Thus, health data derived from population based nationwide health registers in Denmark are representable for the whole Danish population regardless of socioeconomic level.

### 4.4. Limitations

This is a register-based study which has several limitations. The combined data from the registers allowed to adjust for numerous health related conditions. However, clinical data, i.e., smoking status, lifestyle measurements, cognition, education, and functional status were not available. Former studies suggest that cognitive decline combined with premorbid educational underachievement is associated with increased risk for functional decline [45]. Thus, these adverse conditions would be expected to contribute additionally to the risk of infection. Additionally, the paucity of the data from prescriptions and diagnoses from the primary sector is expected to be associated with the incomplete information about the real health status of the population. Nevertheless, access to outpatient health data in this study gave us the possibility to gather information about milder forms of physical illness and therefore provide more robust estimates. However, the possibility for bias associated with the differences in healthcare utilization between individuals with schizophrenia and general population needs to be considered. Existing literature suggest that patients with schizophrenia avoid a contact with the healthcare system until their problems are severe and require hospital admission [46]. Thus, the possibility of underestimation of medical conditions in schizophrenia exists.

Finally, a major difficulty allied to observational studies lies in the definition of onset of the illness. OPUS trial reported that patients with schizophrenia had a median duration of untreated psychosis of approximately 50 weeks before being diagnosed [47]. Consequently, the index date for this subgroup could be falsely delayed with respect to the time of diagnosis. However, this delay could not explain the uniformity of the associations in the whole period of the study.

## 5. Conclusions

To summarize, we found that schizophrenia patients had a greater frequency of almost all types of hospital registered infections, both before and after the diagnosis. Additionally, findings from this study implicate that physical illness plays an important role in the susceptibility to infections in schizophrenia patients as it increased the risk almost three times. We could not, however, confirm the theory that the time after schizophrenia onset were associated with a greater risk of infections. This might be a result of many possible factors that have not been captured in this study due to the limitations of the study design. Identification of potential risk factors and understanding of their role in liability to infection in schizophrenia is essential in the context of risk assessment and prevention of infectious diseases in schizophrenia. Thus, future studies in this field are still needed and they should consider both the heterogeneity of the schizophrenia population and the diversity of conditions occurring in schizophrenia which can modify the risk of infection.

## Figures and Tables

**Figure 1 jcm-07-00485-f001:**
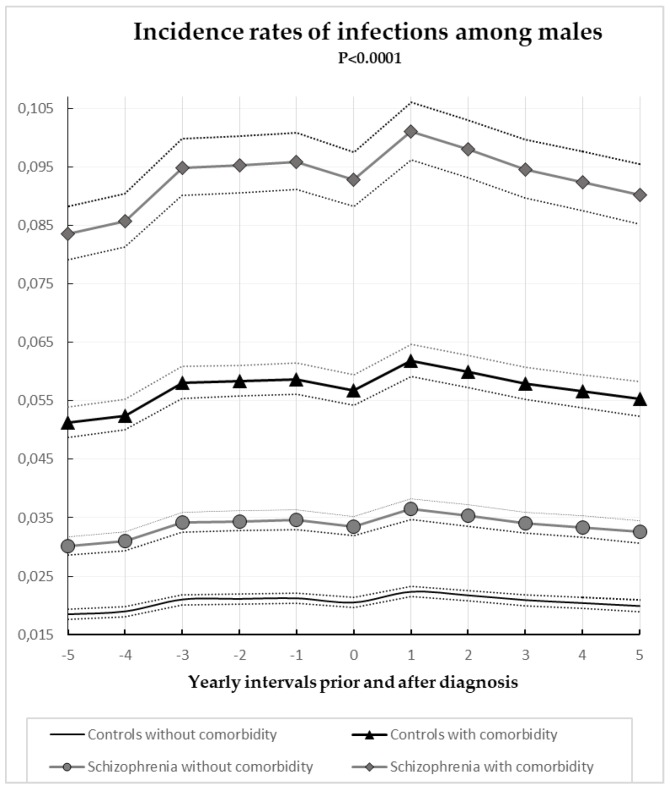
Annual incidence rates of infections for males divided by schizophrenia subjects, controls, and comorbidity 95% confidence intervals.

**Figure 2 jcm-07-00485-f002:**
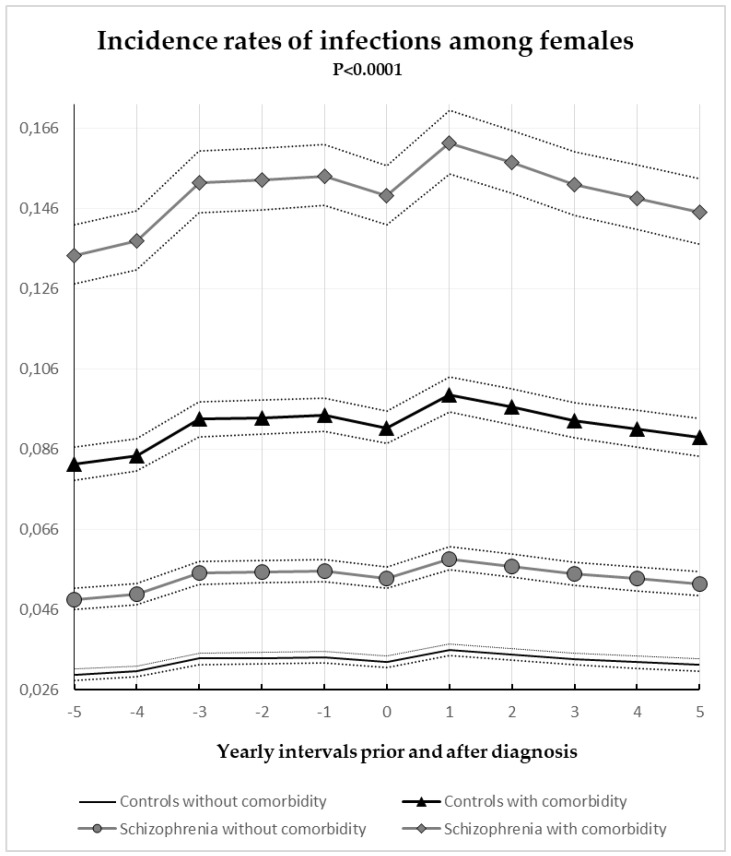
Annual incidence rates of infection for females divided by schizophrenia subjects, controls, and comorbidity 95% confidence intervals.

**Table 1 jcm-07-00485-t001:** Characteristics of the study population. Data derived from study cohort of 86,372 persons born between 1975 and 1990.

Characteristics of the Study Population	Schizophrenia *N* (%)	Control *N* (%)	Total *N* (%)	*p* Value
Year of Birth				
1975–1979	2516	25,160	27,676	
1980–1985	2784	27,840	30,624	
1986–1990	2552	25,520	28,072	
Gender				
Male	4772	47,720	52,492	
Female	3080	30,800	33,880	
Comorbidity Component				
Myocardial Infarction	13 (0.17%)	25 (0.03%)	38 (0.04%)	<0.0001
Congestive Heart Failure	10 (0.13%)	37 (0.05%)	47 (0.05%)	0.0003
Peripheral Vascular Disease	18 (0.23%)	95 (0.12%)	113 (0.13%)	0.02
Cerebrovascular Disease	46 (0.59%)	178 (0.23%)	224 (0.26%)	<0.0001
Dementia	4 (0.05%)	7 (0.01%)	11 (0.01%)	0.01
Chronic Pulmonary Diseases	434 (5.53%)	2125 (2.71%)	2539 (2.96%)	<0.0001
Connective Tissue Disease	43 (0.55%)	325 (0.41%)	368 (0.43%)	0.09
Peptic Ulcer Disease	67 (0.85%)	179 (0.23%)	246 (0.28%)	<0.0001
Diabetes Mellitus	134 (1.71%)	523 (0.67%)	657 (0.76%)	<0.0001
Moderate to Severe Chronic Kidney Disease	43 (0.55%)	230 (0.29%)	273 (0.32%)	0.0004
Hemiplegia	19 (0.24%)	141 (0.18%)	160 (0.19%)	ns
Cancer	44 (0.56%)	476 (0.61%)	520 (0.60%)	ns
Liver Disease	44 (0.56%)	79 (0.10%)	123 (0.14%)	<0.0001
Obesity	308 (3.92%)	1556 (1.98%)	1864 (2.16%)	<0.0001
Substance and Alcohol Abuse and Alcohol Related Diseases	1613 (20.54%)	2744 (3.49%)	4357 (5.04%)	<0.0001
Inflammatory Bowel Syndrome	152 (1.94%)	929 (1.18%)	1081 (1.25%)	<0.0001
Pancreatitis	27 (0.34%)	85 (0.11%)	112 (0.13%)	<0.0001

ns: non-significant.

**Table 2 jcm-07-00485-t002:** Risk of major infections in schizophrenia persons, stratified by type of infection. Data derived from study cohort of 893,647 persons born between 1975 and 1990.

Type of Infection	Incidence Rate Ratios (95% CI)	
	Basic Model *	Adjusted Model **
Any infection ***	1.99 (1.93–2.06)	1.63 (1.58–1.68)
Sepsis ^a^	2.32 (1.86–2.88)	1.34 (1.07–1.66)
Gastrointestinal	1.49 (1.39–1.59)	1.24 (1.17–1.33)
Skin	2.42 (2.32–2.52)	1.88 (1.81–1.96)
Respiratory	1.92 (1.81–2.03)	1.43 (1.35–1.52)
Urological	1.97 (1.82–2.14)	1.80 (1.65–1.95)
Genital	1.89 (1.71–2.08)	1.91 (1.73–2.11)
Otitis media	1.41 (1.27–1.56)	1.29 (1.17–1.43)
Tuberculosis	3.15 (2.27–4.36)	2.35 (1.69–3.27)
CNS Infections ^b^	1.20 (0.96–1.51)	0.98 (0.78–1.23)
Other Types	1.31 (1.23–1.38)	1.17 (1.10–1.23)

* adjusted for calendar year, *p* < 0.0001; ** adjusted for comorbidity, sex, calendar year, *p* < 0.0001; *** all types of infection; ^a^
*p* = 0.008; ^b^ CNS Infections—Infections of the central nervous system, *p* = 0.904.

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
