# Peer review of "Incidence Rates and Risk of Hospital Registered Infections among Schizophrenia Patients before and after Onset of Illness: A Population-Based Nationwide Register Study"

_jcm, 2018, doi:10.3390/jcm7120485_

Round 1
Reviewer 1 Report
This paper has errors in grammar, punctuation, spelling, word-choice and clarity.
The focus of the paper is somewhat redundant with many prior studies, and the findings are not surprising. The sample sizes presented in the table (e.g., over 893,647 controls) do not match the number given in the text (78,520). The statement that all outpatient contacts are included, in the Methods section, does not match the statement later that the focus is on specialist contacts. The description of the table results in the text does not match the table (e.g., most common condition being chronic pulmonary disease, at 0.05% much lower than the rate of 5.53% for Connective Tissue disease. A strange puzzle is the low incidence of cancer, which replicates many earlier studies, but is not mentioned. The figure is difficult to read because the dashed lines are not distinct enough, and crowded by smaller unidentified lines (possibly the confidence intervals, but not explained). The genetic studies are purported to link schizophrenia to the immune system, but they are weak in that regard. The overall finding of the study is not surprising and not crucially informative-- it is well known that persons with schizophrenia need more careful medical care.
Author Response
1) This paper has errors in grammar, punctuation, spelling, word-choice, and clarity.
Thank you very much for this comment. We have addressed this concern by conducting the proofreading of this paper by the extern professional.
2) The focus of the paper is somewhat redundant with many prior studies, and the findings are not surprising.
We do appreciate this observation. The design and the sample of the study are indeed similar to the previous Danish study conducted by Nielson et al. We add to the evidence the new information about occurrence and risk of several types of infections in the periods surrounding the first registration with the schizophrenia diagnosis, and we adjust for several chronical illnesses.
3) The sample sizes presented in the table (e.g., over 893,647 controls) do not match the number given in the text (78,520).
Thank you for this comment. The previous demographic regarded the whole general population. We have replaced these with the matched controls. The changes in the table are highlighted with the yellow color.
4) The statement that all outpatient contacts are included, in the Methods section, does not match the statement later that the focus is on specialist contacts.
We are grateful for this observation. Indeed, these two statements were not clear. We have clarified the sentence in the Strengths –section. The changes in the text are highlighted with the yellow color.
5) The description of the table results in the text does not match the table (e.g., the most common condition being chronic pulmonary disease, at 0.05% much lower than the rate of 5.53% for Connective Tissue disease.
Thank you again for this valuable observation. It is a major deficiency that description of the table is with a disagreement of the explanation in the text. Unfortunately, it happened due to a silly technic error. During the process of copy and paste and resizing the table to the new Word template – the rows in the table moved downwards resulting in the wrong numbers. Thank you for capturing this mistake. We made a new table in Excel avoiding possible errors related to the compatibility of different types of Word. According to the table, Chronic pulmonary diseases with the prevalence of 5.53% are more frequent than Connective Tissue disease with a rate of 0.55%.
6) A strange puzzle is the low incidence of cancer, which replicates many earlier studies, but is not mentioned.
Thank you very much for this comment. We have added a short sentence about this finding with the appropriate reference.
7) The figure is difficult to read because the dashed lines are not distinct enough and crowded by smaller unidentified lines (possibly the confidence intervals, but not explained).
Thank you for this valuable observation. We have improved our figures, and we added 95% confidence intervals to the description of the data.
8) The genetic studies are purported to link schizophrenia to the immune system, but they are weak in that regard.
Thank you for this comment. We have added the sentence about the lack of evidence supporting the genetic influence on the association between schizophrenia and infection. Section 4.2.-Time and risk of infections. The line highlighted with the yellow color.
9) The overall finding of the study is not surprising and not crucially informative-- it is well known that persons with schizophrenia need more careful medical care.
We are thankful for this concern. It is well acknowledged that persons with schizophrenia die almost two decades earlier than persons from the general population due to preventive causes. One of those causes is infections. In contrast to the quite extensively researched field about mortality in schizophrenia, there is a shortage of data about the frequency of infections in schizophrenia PMID: 17919153
Lack of this data makes it difficult to establish whether the high rates of mortality due to infectious causes in schizophrenia are the reflection of more extensive exposure to the infections in this population or whether they reflect higher rates of unfavorable outcomes due to the disease in schizophrenia.
The preventive strategies would have quite a different direction depending on the cause. Thus, we do believe that even if the results are not surprising – the information provided by this study is essential concerning an urgent need for preventive efforts. To the best of our knowledge, this is the first study that provides detailed knowledge about the risk of several different types of infections in a young population of schizophrenia.

Reviewer 2 Report
Re jcm-386596
In their manuscript, Pankiewicz-Dulacz and co-authors, explore the prevalence and risk for infections among individuals before and after a diagnosis of schizophrenia. The authors also investigate the impact of chronic co-morbidities on the risk of infection.
This study is conducted in Denmark and uses Danish population-based registers to assess registered diagnoses involving infections, co-morbidities and schizophrenia. A nested case-control approach is used to identify cases and 10x comparison individuals matched on sex and age in the general population.
This is a clinically important study that suggests that individuals with schizophrenia are at increased risk for contracting any kind of infection at any time point before and during the course of their illness.
I have the following suggestions and questions for the authors:
1. A number of studies have reported on the association between infections and schizophrenia. In the introduction (line37-40), the authors provide, what appears to be a somewhat arbitrary, number of references mixing original studies with reviews and meta-analyses. I would prefer either a more comprehensive list of original studies. Alternatively indicate the use of only up-to-date reviews.
2. As noted by the authors, it has not been established if infections are causally involved in schizophrenia or are associated with the genetic/familial predisposition to develop schizophrenia with no causal role in the development/progression of disease. A few recent studies have addressed the familial component (e.g. PMID 29450471) as well as the polygenetic risk for developing schizophrenia (PMID 27364036) and find that these factors do not explain the association between infections and schizophrenia.
3. The authors chose to study the prevalence of infections five years before and after diagnosis but do not provide any apparent rationale for why five. Perhaps the authors can add one sentence on this?
4. Were co-morbidities diagnosed at any time point during the period 1995-2013, also outside of the +/- five years? Is it possible that some of these were consequences of infections rather than contributing to increased susceptibility to infections? The authors suggest that comorbidity is an important predictor of infections (lines 200-201) in both the general population an in the population of individuals diagnosed with schizophrenia.
5. The authors discuss prior studies of immune related abnormalities in schizophrenia (lines 283-287). The authors could mention a number of studies that have reported on immune system abnormalities well before onset of schizophrenia and other non-affective psychoses (e.g. PMIDs 25152432, 23423137, 29249827).
6. Would it be possible and within the scope of the present study to take the potential confounding by familial factors into account in the present study, as discussed (lines 288-290)?
Author Response
In their manuscript, Pankiewicz-Dulacz and co-authors, explore the prevalence and risk for infections among individuals before and after a diagnosis of schizophrenia. The authors also investigate the impact of chronic co-morbidities on the risk of infection.
This study is conducted in Denmark and uses Danish population-based registers to assess registered diagnoses involving infections, co-morbidities, and schizophrenia. A nested case-control approach is used to identify cases and 10x comparison individuals matched on sex and age in the general population.
This is a clinically important study that suggests that individuals with schizophrenia are at increased risk for contracting any kind of infection at any time point before and during the course of their illness.
I have the following suggestions and questions for the authors:
1. A number of studies have reported on the association between infections and schizophrenia. In the introduction (line37-40), the authors provide, what appears to be a somewhat arbitrary, number of references mixing original studies with reviews and meta-analyses. I would prefer either a more comprehensive list of original studies. Alternatively, indicate the use of only up-to-date reviews.
Thank you very much for this valuable suggestion. The references in the introduction (line 37-40) have been replaced by the number of comprehensive original studies.
2. As noted by the authors, it has not been established if infections are causally involved in schizophrenia or are associated with the genetic/familial predisposition to develop schizophrenia with no causal role in the development/progression of the disease. A few recent studies have addressed the familial component (e.g., PMID 29450471) as well as the polygenetic risk for developing schizophrenia (PMID 27364036) and find that these factors do not explain the association between infections and schizophrenia.
3.
We are deeply thankful for this very relevant comment. As mentioned in the study, an association between infections and schizophrenia might be mediated by several factors, and at the moment no plausible explanation exists that could fully explain this association. Despite growing evidence linking schizophrenia with the immune system, the data is still scarce, and some of the findings are somewhat inconsistent. For instance, the Swedish population-based study found that familial component contributed significantly to the association between infection and schizophrenia. However, the recently published Swedish study could not confirm this hypothesis (PMID 29450471). Similarly, the immune dysfunction in neonates has been linked to the development of the non -affective psychosis. However, findings for the Danish study could not support this theory. Our understanding of how the infections or different pathogens impact the immune system is not elucidated. The results regarding the site of infections and their impact on the development of the non-affective psychosis are also somewhat inconsistent. (association between bacterial contra virus infections with the non-affective psychosis, or CNS infections and their association with the non-affective psychosis).
We have addressed the concerns mentioned above in 2 sentences in section 4.2. The changes in the text are highlighted with the yellow color.
4. The authors chose to study the prevalence of infections five years before and after diagnosis but do not provide any apparent rationale for why five. Perhaps the authors can add one sentence on this?
Thank you very much for this comment. We have added a brief explanation of the choice of the time-frame of follow-up of the study at the end of the Introduction- changes in the text are highlighted with yellow color.
5. Were co-morbidities diagnosed at any time point during the period 1995-2013, also outside of the +/- five years? Is it possible that some of these were consequences of infections rather than contributing to increased susceptibility to infections? The authors suggest that comorbidity is an important predictor of infections (lines 200-201) in both the general population and in the population of individuals diagnosed with schizophrenia.
Thank you very much for this comment. The co-morbidities were diagnosed before the start of follow-up and then at each year during the entire period of the follow-up. We did not explore co-morbidities after that period. We have changed the sentence regarding co-morbidity to make it more clearly for the readers. Method section- text highlighted with yellow color.
We are additionally grateful for the question “Is it possible that some of these were consequences of infections rather than contributing to increased susceptibility to infections?” We think that this is a very interesting query which requires further discussion.
The association between physical illness and infections is complex and not fully understood. Based on the current knowledge we can propose that this association could be bidirectional, at least with regards to some of the illnesses. For instance, some data is indicating that patients with type 2 diabetes have increased rates of antibiotics exposure before the diagnosis PMID: 26312581. On the other hand, diabetes is considered as the risk factors for infections due to such mechanism as decreased T cell-mediated immune response and impaired neutrophil function.
The other example is chronic obstructive pulmonary disease, where data support bidirectional association PMID 20935112. Similarly –some types of lymphomas PMID:22552810. Thus, we cannot rule out the possibility that some of the physical illness measured in this study could be a consequence of prior infections. However, they still would be considered as the risk factor for the development of an infection in the future. Thus, they can be considered as an appropriate and relevant confounder in our study.
6. The authors discuss prior studies of immune-related abnormalities in schizophrenia (lines 283-287). The authors could mention a number of studies that have reported on immune system abnormalities well before the onset of schizophrenia and other non-affective psychoses (e.g., PMIDs 25152432, 23423137, 29249827).
Thank you for this valuable suggestion. We have supplied discussion with suggested data.
7. Would it be possible and within the scope of the present study to take the potential confounding by familial factors into account in the present study, as discussed (lines 288-290)?
Thank you again for this valuable suggestion. We do think that taking potential confounding by familial factors into account would indeed, be crucial for better understanding of the association between schizophrenia and infection. Unfortunately, it is not within the scope of the present study. Still, we consider a more suitable design in our future research to investigate the temporal association between infection and schizophrenia as shown in this study, where familial influences would be taken into account.
